# Serum Adiponectin Levels Increase in Acute Ischemic Stroke and Correlate with Patients’ Outcomes: A Pilot Study

**DOI:** 10.3390/biomedicines12081828

**Published:** 2024-08-12

**Authors:** Andrei-Lucian Zaharia, Violeta Diana Oprea, Camelia Alexandra Coadă, Claudiu Elisei Tănase, Ana-Maria Ionescu, Sergiu Ioachim Chirila, Raul Mihailov, Dana Tutunaru, Mihaiela Lungu

**Affiliations:** 1Faculty of Medicine and Pharmacy, “Dunărea de Jos” University of Galaţi, 800216 Galaţi, Romania; zaharia.andreilucian@gmail.com (A.-L.Z.); diana.v.oprea@gmail.com (V.D.O.); tanaseclaudiumd@gmail.com (C.E.T.); raulmihailov@yahoo.com (R.M.); mihaelalungu17@yahoo.com (M.L.); 2“St. Apostle Andrei” Clinical Emergency County Hospital Galaţi, 800578 Galaţi, Romania; 3Faculty of Medicine, “Iuliu Haţieganu” University of Medicine and Pharmacy, 400012 Cluj-Napoca, Romania; 4Faculty of Medicine, Ovidius University of Constanța, 900470 Constanța, Romania; iuliusana@gmail.com (A.-M.I.); sergiu.chirila@univ-ovidius.ro (S.I.C.)

**Keywords:** adiponectin, *ADIPOQ*, stroke, ischemic, diagnosis

## Abstract

Stroke is a leading cause of death and severe disability worldwide. Rapid diagnosis is critical to ensure the timely administration of medical treatment. Given that in some cases CT scans fail to show the classic clinical signs of stroke, we aimed to evaluate the diagnostic capacity of adiponectin levels and their association with the clinical parameters of patients with acute ischemic stroke (AIS). Adiponectin was measured within 24 h (T1) and 48 h (T2) of AIS onset in 70 patients. A total of 68 control cases were included in the study. Adiponectin levels were significantly higher in the AIS patients than in the controls (16.64 (3.79; 16.69) vs. 3.78 (3.79; 16.69); *p* < 0.001), with an accuracy of 0.98 (AUC = 0.99). Lower levels were seen in males and in AIS patients with obesity. Higher levels of adiponectin at T1 were associated with a moderate/severe NIHSS score at patient discharge. Moreover, higher levels of borderline significance were seen in patients who died within 12 months of their AIS episode (*p* = 0.054). Adiponectin may be a useful biomarker for the identification of AIS patients who do not present classic CT signs and could be used to stratify severe cases. Further studies are needed to validate these results.

## 1. Introduction

Stroke is a leading cause of death and severe disability worldwide. Its rapid progression and resulting irreversible brain damage requires fast medical intervention within the first few hours of onset. The prompt evaluation and diagnosis of stroke patients is a critical step to ensure adequate treatment, which can enhance patients’ recovery and minimize long-term and permanent disability [1].

Extensive research has been dedicated to exploring biomarkers that can aid in the diagnosis, treatment and prognosis of stroke. In certain cases, differentiating between an acute ischemic stroke (AIS) and a pathology that can simulate a stroke can be difficult, especially in cases with non-specific symptoms and in early presentations, i.e., when curative interventions can ensure complete recovery. In such situations, the use of biomarkers can be especially valuable. An ideal stroke biomarker should allow for a differentiation of stroke from stroke mimics with high specificity and sensitivity. Over 150 candidate biomarkers have been evaluated for their potential use in the diagnosis, treatment and even long-term prognosis of stroke [2].

Adiponectin is a cytokine secreted by adipocytes, with pleiotropic actions involved in carbohydrate and lipid metabolism (as well as insulin sensitivity), but also has anti-inflammatory, anti-fibrotic, vasodilator and anti-oxidant effects, and promotes apoptosis in neoplastic cells [3,4]. It also has a protective effect against diabetes, regulating the level of serum blood glucose [5,6] and atherosclerosis [7,8].

Adiponectin has two major receptors: adiponectin-receptor 1 (AdipoR1) and adiponectin-receptor 2 (AdipoR2), as well as a minor T-cadherin receptor [3,9]. AdipoR1 is predominantly expressed in skeletal muscle cells while AdipoR2 is expressed in hepatocytes, but the expression of these receptors has also been shown in other tissues, such as myocardium, macrophages, endothelial cells, lymphocytes, brain tissue and adipose tissue [10,11], as well as in β pancreatic cells, where the level of AdipoR2 expression is equivalent to that of the liver [12]. In 2004, Hug et al. isolated a third receptor, T-cadherin, which is expressed in vascular endothelial cells and smooth muscle tissue. The expression of this receptor correlates with the progression of atherosclerosis [13], and its low expression causes the exacerbation of cardiac hypertrophy during chronic overload with high pressure, thus suggesting the protective role of T-cadherin [14].

Since adiponectin is associated with multiple cardiovascular risk factors, such as dyslipidemia, type II diabetes and hypertension, its association with stroke is predictable [15,16]. Recent studies have shown that decreased adiponectin levels may be indicative of cerebrovascular disease and form part of the response occurring in stroke patients [17]. Several studies have shown a negative correlation with cardiovascular disease [18,19], while a meta-analysis of prospective studies on the value of adiponectin could not sufficiently prove its correlation with the occurrence of stroke [20]. Thus, low values increase the risk of mortality within five years post-stroke [21] and correlate with the risk of hypertension and cardiomyopathy in diabetic patients [22,23,24]. The protective effects of adiponectin in stroke may also be due to the synthesis of nitric oxide at the level of endothelial cells through the AdipoR1 receptor [25,26,27]. In acute stroke, vasospasm during the first few days to weeks can lead to increased morbidity and mortality. Nitric oxide has a protective role through vasodilation and increased blood flow as its synthesis in endothelial cells is stimulated by adiponectin, thus exerting a possible post-stroke neuroprotective effect [26,28]. Adiponectin serum values are elevated in heart failure, though evidence is still needed for adiponectin as a marker of cardiovascular disease [29]. There are also differences in adiponectin levels by gender, with females having higher values than men. Hormonal changes due to menopause, ovariectomy or estrogen replacement therapy do not seem to alter adiponectin levels [30,31,32,33].

Starting from the body of evidence connecting adipose tissue, adipokines and vascular disorders, this study aimed to investigate the diagnostic capacity of adiponectin in patients presenting with AIS at emergency units. The second aim was to evaluate the association of adiponectin levels with the clinical parameters of AIS as well as the prognosis of these patients. Moreover, we assessed the evolution of adiponectin levels in patients within the first two days from the onset of AIS.

## 2. Materials and Methods

### 2.1. Study Design

This analytical, prospective, single-center study was approved by the Hospital Ethics Committee of “St. Apostle Andrei” Clinical Emergency County Hospital in Galati, Romania, as Decision 524/07.01.2021. The study was conducted according to all the local ethical guidelines for research on humans and the 1964 Helsinki Declaration.

### 2.2. Patients’ Inclusion/Exclusion Criteria and Patients’ Data Collection

We included all consecutive patients admitted at the Emergency Clinical Hospital of the Galati Clinical Neurological Department presenting with clinical signs and symptoms relevant to AIS in the 24 h previous to hospital admittance.

The following specific inclusion criteria for AIS patients were used: a clinical profile suggestive of AIS associated with a native cerebral CT scan performed within the first 24 h after admittance; a CT ASPECTS score of 10 (i.e., no visible indication of AIS modifications) [16]; age of at least 18 years old; the presentation of written informed consent signed by either the patient or a close family member. The exclusion criteria were as follows: patients with other neurological conditions and/or hemorrhagic stroke; concomitant comorbidities such as infectious diseases, acute myocardial infarction, recent surgery or significant traumatic events (within the past 30 days); severe organ deficiencies (renal/hepatic); and a previous cancer diagnosis. For the control group, we used similar exclusion criteria, and the inclusion criterion was the following: patients presenting at the same hospital for other health-related issues without any cerebral ischemic disorders. The period of enrollment for all patients was between January 2022 and May 2023.

We gathered the demographic data, medical history and detailed AIS-related clinical data from each patient or their family members. We used the National Institutes of Health Stroke Scale (NIHSS), evaluated by a certified neurologist, for all patients for the following 3 different stages: at the time of their presentation at the hospital, at 48 h after their admittance and at the time of their discharge from the hospital. We followed the criteria of the Trial of Org 10,172 in the Acute Stroke Treatment (TOAST) classification system [34] in our evaluation of brain CT scans to classify the subtypes of AIS.

### 2.3. Blood Work and Adiponectin Measuring

During the routine blood screening, a BD-vacutainer serum tube without anticoagulants was collected to measure the dosage of adiponectin at presentation (T1) and after 48 h (T2). Samples were frozen at −20 °C until analysis. Serum adiponectin levels were measured using the ELISA method, using a human adiponectin ELISA kit (catalog number E-EL-H6122, provided by Elabscience, Houston, Texas, United States). Briefly, the micro-ELISA plate provided in the kit was pre-coated with an antibody specific to human adiponectin. A biotinylated detection antibody specific to human adiponectin and an Avidin-Horseradish Peroxidase (HRP) conjugate were added to each micro-plate well and incubated. Unbound antibodies were washed, and a substrate solution was then added to each well. Wells containing human adiponectin from the samples shifted to a blue color. The enzyme–substrate reaction was terminated by the addition of a stop solution and the color turned yellow. The optical density (OD) was measured spectrophotometrically at a wavelength of 450 nm ± 2 nm. The concentration of adiponectin in the samples was calculated by comparing the OD of the samples to the standard curve provided by the manufacturer.

### 2.4. Statistical Analysis

Statistical analysis was performed in R [35] version 4.4.0 (2024-04-24 ucrt)—“Puppy Cup”. Continuous variables were reported as the median and interquartile range (IQR), while categorical variables were reported as frequency and percentage. The normality of the continuous variables was tested using the Kolmogorov–Smirnov test, while the equality of variances was tested using the F test or Levene test, as appropriate. For the continuous variables, the differences between the two groups were analyzed using the *t*-test for equal or unequal variances or the Mann–Whitney U test, based on the assumptions of each test. The differences between the three groups were assessed using ANOVA tests or Kruskal–Wallis tests. Categorical variables were tested using the Chi-Square test. The performance of the binary classification model, using adiponectin as a diagnostic biomarker, was assessed through an ROC curve. The Youden index was used to establish the best discriminative threshold. Accuracy, sensitivity, specificity, negative predictive value (NPV) and positive predictive value (PPV) parameters were computed. All tests were two-sided. A *p*-value of <0.05 was considered statistically significant. Logistic regression was used for the univariable analysis of relevant parameters’ impact on the death of AIS patients within 12 months after their episode. Significant variables were combined using the multivariable model.

## 3. Results

### 3.1. Study Population

This study included 70 patients diagnosed with AIS and 68 control patients with other associated pathologies, but without a history of stroke. The AIS patients presented at the emergency room of the “Sf. Ap. Andrei” Emergency Clinical Hospital in Galați and were admitted to the Neurology Clinic for clinical signs relevant to AIS. The general characteristics of the study group and the control group are represented in Table 1.

### 3.2. Serum Adiponectin Levels Increase in AIS Patients

A comparative analysis between AIS patients and non-stroke controls revealed a significant increase in serum adiponectin levels in AIS cases. Specifically, AIS patients exhibited a median adiponectin level of 16.64 ng/mL with a higher variance, while the control group presented a significantly lower level of 3.78 ng/mL (*p*-value < 0.001) (Figure 1A). Adiponectin levels did not change significantly between the two measuring timepoints, showing only a slight decrease of 0.34 ng/mL (*p* = 0.490).

An ROC analysis was performed to evaluate the efficacy of adiponectin to identify AIS patients. The analysis indicated that at a serum threshold of 7.32 ng/mL (Figure 1B), adiponectin demonstrated high accuracy in diagnosing AIS, achieving an accuracy of 0.99, a specificity of 0.99 and a sensitivity of 0.97. The NPV was 0.97, while the PPV was 0.99. The standard error was 0.007 and the 95% confidence interval was from 0.957 to 1. The z statics of 72.69 and the significance level (*p* < 0.001) confirm the robustness of this test.

### 3.3. Serum Adiponectin Levels and Clinical Features of AIS

As presented in Table 2, the following aspects can be highlighted: the largest number of patients, 43 (61.43%), had atherothrombotic AIS, while 27 (38.57%) of the patients had cardioembolic stroke. From the point of view of the affected territory, 32 (45.71%) of the cases were in the territory of the left middle cerebral artery, 24 (34.29%) cases were in the territory of the right middle cerebral artery and 14 (20%) cases were in the vertebrobasilar territory (Table 2). A total of 10 (14.28%) AIS patients presented within a maximum of 4.5 h from the onset of clinical signs and were treated by intravenous thrombolysis with alteplase. A total of eight (11.43%) patients died during hospitalization and another four patients died in the following 12 months. From the point of view of stroke severity at presentation, there were 22 (31.43%) patients with an NIHSS score between 0 and 4 points, 28 (40%) patients with an NIHSS score between 5 and 15 points and 20 (28.57%) patients with an NIHSS score > 16 points (Table 2).

Adiponectin levels were significantly lower in AIS patients with obesity at both timepoints (T1: *p* = 0.034; T2: *p* = 0.038). In terms of stroke severity, higher levels of adiponectin at T1 were associated with moderate/severe stroke at discharge (20.25 (17.11; 22.14)) for patients with an NIHSS score ≥16, *p* = 0.048. No other significant associations were seen at the other timepoints.

### 3.4. Adiponectin and Prognosis of AIS Patients

Within one year of initial AIS episode diagnosis, 12 (17.14%) patients died. We investigated whether adiponectin levels were correlated with this outcome. Median adiponectin concentrations were lower in patients who survived their acute ischemic stroke (AIS) episode, with values of 15.77 ng/mL at T1 and 15.45 ng/mL at T2, with a borderline significance of *p* = 0.054 and *p* = 0.075, respectively (Figure 2A,B). ROC analyses revealed a modest performance of adiponectin in predicting this outcome at both timepoints. Nevertheless, adiponectin had high NPVs.

We sought to evaluate which parameters could be significantly associated with patients’ *exitus* within 12 months after their AIS episode. The univariable analysis revealed grade III blood hypertension, NIHSS at presentation and adiponectin T1 levels above 13.92 ng/mL to be significantly associated with patient death, with *p*-values of 0.031, 0.001 and 0.049, respectively (Table 3). In the multivariable model, blood hypertension and NIHSS maintained their significance as independent risk factors (*p* = 0.039 and 0.005), while adiponectin levels reached a borderline statistical value (*p* = 0.060).

## 4. Discussion

Adiponectin was investigated in the case of many pathologies such as obesity, dyslipidemia, diabetes, hypertension and metabolic dysfunctions [9]. Serum levels were also shown to be altered in neurological diseases involving a pathogenic, metabolic or inflammatory mechanism [9]. Adiponectin receptors are well expressed in many regions of the central nervous system, where they exert neuroprotective and antidepressant effects [9]. In the brain, adiponectin seems to play a role not only in energy homeostasis, but also in neurogenesis and synaptic plasticity, as evidenced by studies on mice [36,37,38].

### 4.1. Adiponectin and Cardiovascular Diseases

Circulating adiponectin inhibits the adhesion of monocytes to endothelial cells and the transformation of macrophages into foam cells by reducing the binding and absorption of LDL-cholesterol [39,40,41]. This process is crucial to atherosclerosis and stroke pathogenesis [42]. At the cardiovascular level, it has a protective effect through several mechanisms, described as follows: it increases the expression of CD36, the absorption and oxidation of fatty acids and the stimulation of insulin-dependent glucose transport, as well as Akt phosphorylation at the level of myocytes [43]. The concentration of serum adiponectin is considered one of the major indicators of atherosclerosis and systemic inflammation. In fact, slightly lower levels were seen in patients with atherothrombotic-type stroke.

### 4.2. Adiponectin and Stroke Onset

Although adiponectin is a major cytokine that affects the pathogenesis of various cardiovascular diseases, its clinical significance in stroke is controversial, with the data on changes in blood adiponectin levels in stroke patients being less consistent. To date, the pathophysiological mechanisms by which serum adiponectin plays a role in the risk and prognosis of stroke are not fully understood [44].

In a study conducted by Wang et al., serum adiponectin levels were higher in the stroke group than in the control group (*p* < 0.001) [45]. However, other studies have shown no difference in adiponectin values between stroke cases and controls [46,47]. The AIS patients from our center showed significantly higher adiponectin levels than the controls. In addition, adiponectin levels in patients with ischemic stroke caused by an atherothrombotic mechanism were significantly lower compared to the control group [48]. In our patients, this tendency was noted but did not reach the threshold for significance. This finding suggests that adiponectin could serve as a tool for the identification of AIS patients, thus allowing them to receive prompt medical care.

### 4.3. Adiponectin and Stroke Prognosis

Nagasawa et al. suggested that elevated plasma adiponectin levels may be a predictor of stroke mortality in the 17 months after an acute episode, while another study confirmed that adiponectin values may help classify stroke subtypes and predict the severity of neurological impairment and functional outcome in patients with ischemic stroke [47,49]. Another study showed that high plasma adiponectin was associated with mortality in patients with established atherosclerosis undergoing surgery for carotid artery stenosis but is not associated with ischemic events [50]. Our results further validate these findings (with borderline significance most likely due to the small number of cases), as the AIS patients who demised within 12 months also had higher adiponectin levels. Nevertheless, in the multivariable analysis aimed at identifying independent predictors for patient death, only blood hypertension and NIHSS at admission remained significant. Due to the small number of such events, we believe that further investigation is warranted to confirm these results and to obtain a more stable and precise model.

As far as the analysis of AIS severity is concerned, adiponectin levels within the first 24 h following stroke onset were associated with an NIHSS score ≥ 16 at discharge. Albeit significant, these results must be interpreted with caution given the small number of cases in this group (*N* = 6). Moreover, this significance was lost at T2. Notably, at this timepoint, not all measurements were possible as some patients died within the first 48 h following admission. This suggests that adiponectin could serve as a prognostic tool for the prediction of long-lasting ischemic damage, should these results be validated.

Joanna Pera et al. reported that in patients with ischemic stroke, plasma levels of adiponectin systematically decreased in the acute phase [51]. This was consistent with the results obtained in an animal model of cerebral ischemia [52]. Sasaki et al. observed a transient reduction in serum levels during acute stroke on the first day of hospitalization compared to day 14 after admission, but nevertheless the exact relationship between the time of measurement and the timing of stroke onset is unclear [53]. In the case of our patients, the decrease in adiponectin levels from T1 (24 h) to T2 (48 h) was insignificant. This suggests that within this timeframe, adiponectin could indeed serve as a tool for aiding the diagnosis and prognosis prediction of AIS patients.

### 4.4. Study Limitations and Future Directions

This study has several limitations. First of all, we only measured the levels of adiponectin in the acute phase of AIS, which might not fully illustrate its variations during the evolution of this disease. However, blood adipokine levels could remain stable over time [54]. The validation of the ELISA-measured adiponectin was questioned by Bluher et al. [55], who reported significant differences between different commercially available tests. In this study, we measured the total levels of adiponectin and not its isoforms. Also, we should mention the relatively small sample size of our AIS cases (*N* = 70), which limited the power of the subgroup analyses, as shown by the wide confidence intervals obtained in the univariable and multivariable analyses. Moreover, the group of patients is composed only of Romanian patients and the analysis comes from a single center. Larger studies, with diverse populations, are needed to further dissect the role of adiponectin in the prognosis of cerebrovascular diseases.

## 5. Conclusions

In conclusion, the results of this study show that serum adiponectin levels were increased in the first 24 and 48 h following the onset of AIS in patients without classic CT-scan signs. Adiponectin demonstrated high accuracy in diagnosing AIS, and higher levels were borderline-associated with patients’ demise within 12 months of their AIS episode.

## Figures and Tables

**Figure 1 biomedicines-12-01828-f001:**
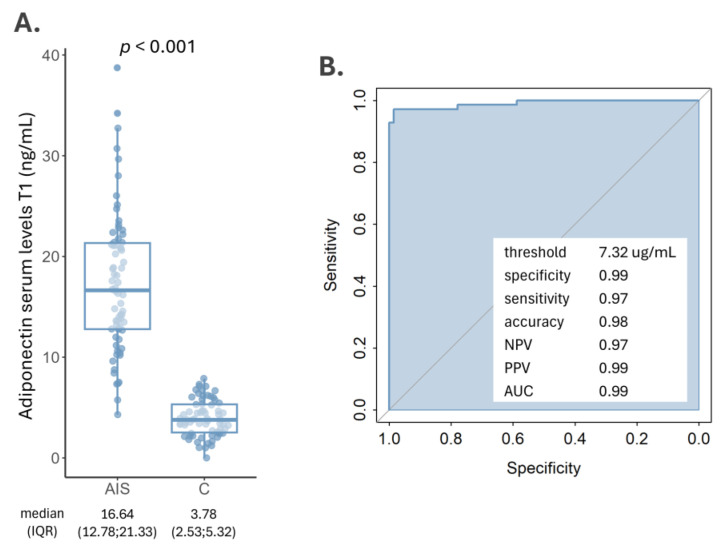
(**A**) Adiponectin levels in AIS patients and controls; (**B**) ROC curve (direction controls < AIS cases) showing the discriminatory capacity between AIS cases and controls of adiponectin serum levels within 24 h after stroke onset at a threshold of 7.32 ng/mL. AIS: acute ischemic stroke; C: control; NPV: negative predictive value; PPV: positive predictive value; AUC: area under the curve.

**Figure 2 biomedicines-12-01828-f002:**
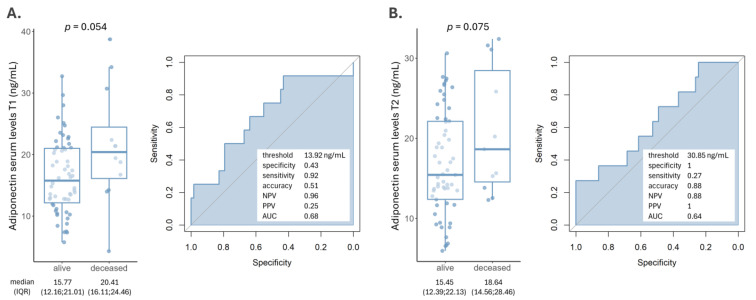
Adiponectin levels at T1-24 h (**A**) and T2-48 h (**B**) between patients who died within a year of their AIS event and those who survived. ROC analysis with the following direction: controls < AIS cases revealed a modest performance but high NPVs.

**Table 1 biomedicines-12-01828-t001:** Demographic, clinical and pathological profiles of patients diagnosed with acute stroke and control group.

Variable		Control Group *N* = 68	AIS Group *N* = 70	*p*-Value
Age (years), *mean ± sd*		69.28 ± 10.5	70.24 ± 10.93	0.598
Sex, *N (%)*	Female	30 (44.12)	27 (38.57)	0.508
	Male	38 (55.88)	43 (61.43)	
Residential background, *N (%)*	Urban	36 (52.94)	46 (65.71)	0.127
	Rural	32 (47.06)	24 (34.29)	
Atrial fibrillation, *N (%)*		17 (25)	27 (38.57)	0.087
Dyslipidemia, *N (%)*		34 (50)	43 (61.43)	0.177
Diabetes, *N (%)*		27 (39.71)	15 (21.43)	0.020
Hypertension, *N (%)*	grade 1	11 (16.18)	5 (7.14)	0.027
	grade 2	32 (47.06)	24 (34.29)	
	grade 3	25 (36.76)	41 (58.57)	
Chronic alcohol use, *N (%)*		21 (30.88)	10 (14.29)	0.020
Chronic smoker, *N (%)*		23 (33.82)	10 (14.29)	0.007

*N*: number of patients; AIS: acute ischemic stroke; sd: standard deviation.

**Table 2 biomedicines-12-01828-t002:** Adiponectin levels based on the clinical characteristics of the AIS patients included in this study.

Variable		*N (%)*	Adiponectin T1 Median (IQR)	*p*-Value	Adiponectin T2 Median (IQR)	*p*-Value
Sex	Female	27 (38.57)	20.25 (15.55; 21.88)	0.030	18.21 (14.32; 23.43)	0.146
	Male	43 (61.43)	14.21 (11.58; 20.13)		14.15 (11.66; 20.13)	
Diabetes	No	53 (78.47)	16.55 (12.73; 21.28)	0.442	16.04 (12.55; 22.54)	0.942
	Yes	15 (21.43)	18.77 (13.34; 21.6)		15.53 (13.76; 20.98)	
Obesity (BMI ≥ 30 [kg/m^2^])	No	59 (86.76)	17.42 (12.98; 21.75)	0.034	16.98 (13.13; 24.02)	0.038
	Yes	9 (13.24)	12.8 (8.42; 16.86)		14.22 (9.39; 15.65)	
Chronic alcohol use	No	70 (85.71)	16.44 (12.52; 21.08)	0.280	15.37 (12.43; 22.34)	0.375
	Yes	10 (14.29)	20.14 (13.3; 22.74)		18.06 (14.35; 24.1)	
Chronic smoker	No	70 (85.71)	16.8 (12.94; 21.23)	0.494	16.04 (13.6; 22.47)	0.468
	Yes	10 (14.29)	12.53 (9.76; 22.42)		13.49 (10.54; 20.97)	
Hypertension	grade 1	5 (7.14)	15.34 (11.77; 16.33)	0.771	14.31 (11.89; 26.67)	0.537
	grade 2	24 (34.29)	17.91 (10.74; 21.08)		15.02 (12.21; 20.12)	
	grade 3	41 (58.57)	16.74 (12.98; 21.4)		17.48 (13.72; 23.15)	
Stroke type	Cardioembolic	27 (38.57)	18.23 (13.55; 21.89)	0.325	19.14 (14.53; 25.19)	0.034
	Atherothrombotic	43 (61.43)	16.33 (11.47; 20.81)		14.27 (11.75; 20.68)	
Affected vascular territory	Left middle cerebral artery	32 (45.71)	16.64 (11.62; 21.23)	0.368	15.48 (12.61; 21.84)	0.634
	Right middle cerebral artery	24 (34.29)	18.81 (14.03; 21.65)		18.06 (13.74; 24.03)	
	Vertebrobasilar system	14 (20)	14.21 (12.78; 16.9)		13.89 (12.48; 24.5)	
Received thrombolysis	No	60 (85.72)	16.46 (12.51; 21.1)	0.104	15.59 (12.61; 22.34)	0.382
	Yes	10 (14.28)	18.82 (14.74; 22.5)		16.96 (13.49; 25.84)	
NIHSS severity						
At presentation	minor (NIHSS = 0–4)	22 (31.43)	17.14 (12.78; 21.6)	0.098	14.86 (12.48; 26.27)	0.589
	moderate (NIHSS = 5–15)	28 (40)	13.93 (11.54; 20.34)		15.53 (12.22; 21.52)	
	moderate to severe (NIHSS ≥ 16)	20 (28.57)	19.1 (15.8; 22.5)		16.04 (13.66; 24.88)	
At 48 h	no stroke signs (NIHSS = 0)	2 (2.86)	17 (15.1; 18.89)	0.520	14.42 (14.06; 14.78)	0.834
	minor (NIHSS = 0–4)	26 (37.14)	16.1 (11.14; 21.6)		14.59 (11.7; 25.93)	
	moderate (NIHSS = 5–15)	27 (38.57)	14.8 (12.28; 20.72)		15.53 (13.02; 21.52)	
	moderate to severe (NIHSS ≥ 16)	15 (21.43)	19.43 (15.47; 21.89)		16.51 (14.2; 23.26)	
At discharge	no stroke signs (NIHSS = 0)	12 (19.35)	12.99 (11.04; 14.75)	0.048	13.24 (9.45; 16.07)	0.074
	minor (NIHSS = 0–4)	27 (43.55)	18.9 (12.8; 22.89)		19.93 (14.25; 26.27)	
	moderate (NIHSS = 5–15)	17 (27.42)	14.57 (12.98; 16.55)		13.99 (12.55; 19.39)	
	moderate to severe (NIHSS ≥ 16)	6 (9.67)	20.25 (17.11; 22.14)		16.98 (15.3; 24.27)	

*N*: number of patients; IQR: interquartile range; BMI: body mass index; NIHSS: National Institutes of Health Stroke Scale; h: hours.

**Table 3 biomedicines-12-01828-t003:** Univariable and multivariable analysis of parameters impacting AIS patients’ death within 12 months of their stroke episode.

		Univariable Analysis		Multivariable Analysis	
Variable		OR (95%CI)	*p*-Value	OR (95%CI)	*p*-Value
Sex	ref = female	0.86 (0.24; 3.2)	0.809		
Age		1.05 (0.99; 1.12)	0.149		
Obesity	ref = normal weight	0.57 (0.03; 3.59)	0.611		
Chronic smoker	ref = non-smoker	0.49 (0.03; 3.06)	0.525		
Chronic alcohol use	ref = no use	1.25 (0.17; 5.98)	0.796		
Atrial fibrillation	ref = absent	2.66 (0.75; 10.03)	0.131		
Diabetes	ref = absent	2.14 (0.5; 8.19)	0.277		
Blood hypertension	ref = grade I + II	10.27 (1.24; 84.76)	0.031	19.91 (1.16; 342.6)	0.039
Stroke type	ref = cardioembolic	0.38 (0.1; 1.32)	0.131		
NIHSS at presentation		1.23 (1.09; 1.39)	0.001	1.28 (1.08; 1.53)	0.005
Adiponectin T1	ref ≤ 13.92 ng/mL	8.33 (1.47; 157.37)	0.049	14.08 (0.90; 221.74)	0.060

OR: odds ratio; CI: confidence interval.

## Data Availability

Data can be provided upon reasonable request to the corresponding authors.

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
