# Peer review of "Serum Adiponectin Levels Increase in Acute Ischemic Stroke and Correlate with Patients’ Outcomes: A Pilot Study"

_biomedicines, 2024, doi:10.3390/biomedicines12081828_

Round 1

Reviewer 1 Report

Comments and Suggestions for Authors

This study only contained 70 patients and 68 controls, so the sample is not enough big. All results are based on the univariate analyses, which are not recommonded in these times. 

Comments on the Quality of English Language

The English can be accepted.

Author Response

This study only contained 70 patients and 68 controls, so the sample is not enough big. All results are based on the univariate analyses, which are not recommonded in these times. 

R: Thank you for this excellent suggestion. We performed a univariable and multivariable binary logistic regression analysis which can now be found in table 3. We also added these new results in the text: “We sought to evaluate which parameters could be significantly associated with patients’ exitus within 12 months after the AIS episode. Univariable analysis revealed grade III blood hypertension, NIHSS at presentation and adiponectin T1 levels above 13.92 ng/mL to be significantly associated with patient death, p-values of 0.031; 0.001 and 0.049, respectively (Table 3). In the multivariable model, blood hypertension and NIHSS maintained their significance as independent risk factors (p=0.039 and 0.005) while adiponectin levels reached a borderline statistical value (p=0.060).” (lines 205-210).

As far as the sample size of the study is concerned, we agree that it might seem that the sample size is insufficient. However, we calculated the power of out study for the main aim and obtained a power of 1 for our sample size. This is due to the fact that the effect size of adiponectin levels between the two groups is large (Cohen D = 2.62).

Reviewer 2 Report

Comments and Suggestions for Authors

Overview: In this study, the authors evaluated the diagnostic performance of serum adiponectin levels and their association with the clinical parameters of patients with acute ischemic stroke (AIS). The manuscript was technically well-written and presented results that showed the potential usefulness of adiponectin for AIS screening and prognosis, introducing new data to the field. Although there were some clinical limitations to the study findings, these were presented in the discussion section.

I do have some comments and questions.

1.     In the abstract (line 23): 'Moreover, higher levels were seen in patients who deceased within xxx months from the AIS episode (p=0.054).' Please specify the number of months.

2.     The introduction provides a good overview of the state-of-the-art in the subject of study. However, I suggest the authors improve the fifth paragraph by presenting recent findings more concisely and clearly highlighting the current gap their research aims to address.

3.     Considering the translational purposes of serum adiponectin in AIS, I wonder about the implications of the study design for its clinical applicability. The authors aimed to evaluate the diagnostic capacity of adiponectin levels and their association with the clinical parameters of patients with acute ischemic stroke. However, they did not include patients with similar clinical presentations in the emergency room. Additionally, there is a risk of measuring the diagnostic performance of individual serum biomarkers without accounting for other potential confounding factors or pathologies that could impair serum adiponectin levels, such as renal dysfunction and cancer. Have the authors evaluated the correlation of adiponectin levels with hypertension, chronic alcohol use, and chronic smoking, given the statistical differences observed between groups? Have the authors considered the selectivity of adiponectin for AIS?

4.     The level of adiponectin in the AIS group was highly variable. Have the authors considered evaluating the AIS patients by quartiles according to their serum adiponectin levels?

5.     The discussion section includes extensive references to other studies, which sometimes makes it resemble a review. To enhance discussion, the authors could more clearly state how their results align or differ from previous reports and then discuss the clinical implications of these findings for the study population.

Author Response

Overview: In this study, the authors evaluated the diagnostic performance of serum adiponectin levels and their association with the clinical parameters of patients with acute ischemic stroke (AIS). The manuscript was technically well-written and presented results that showed the potential usefulness of adiponectin for AIS screening and prognosis, introducing new data to the field. Although there were some clinical limitations to the study findings, these were presented in the discussion section.

R: Thank you for taking the time to thoroughly review our manuscript and for all your useful recommendations. We went through all the comments and implemented them. Please see below out point-by-point answers. We feel that the adjustments we made have significantly improved the quality and clarity of our work.

I do have some comments and questions.

  1. In the abstract (line 23): 'Moreover, higher levels were seen in patients who deceased within xxx months from the AIS episode (p=0.054).' Please specify the number of months.

R: Thank you for noticing this typo. We added the number of months (12 months, as mentioned below in the text)

  1. The introduction provides a good overview of the state-of-the-art in the subject of study. However, I suggest the authors improve the fifth paragraph by presenting recent findings more concisely and clearly highlighting the current gap their research aims to address.

R: Thank you for this suggestion on ways to enhance our introduction section. We rearranged the paragraph to address these requirements.

  1. Considering the translational purposes of serum adiponectin in AIS, I wonder about the implications of the study design for its clinical applicability. The authors aimed to evaluate the diagnostic capacity of adiponectin levels and their association with the clinical parameters of patients with acute ischemic stroke. However, they did not include patients with similar clinical presentations in the emergency room. Additionally, there is a risk of measuring the diagnostic performance of individual serum biomarkers without accounting for other potential confounding factors or pathologies that could impair serum adiponectin levels, such as renal dysfunction and cancer. Have the authors evaluated the correlation of adiponectin levels with hypertension, chronic alcohol use, and chronic smoking, given the statistical differences observed between groups? Have the authors considered the selectivity of adiponectin for AIS?

R: Thank you for this comment giving us the opportunity to clarify these points. Indeed, comorbidities such as renal impairment and concomitant tumors could impact our results. For this reason, we excluded patients presenting at our hospital with such pathologies and included only patients suffering from AIS. We did not select patients without metabolic and cardiovascular diseases as these are well known risk factors for AIS thus, we wanted our selected cases to be representative for the general stroke population. Nevertheless, we analyzed the influence of these factors on adiponectin levels as well, so that any potential significance can be taken into account. Please see Table 2 in which we added the analysis of adiponectin levels based hypertension, alcohol consumption and smoking habits, which showed no significant differences between groups.

As far as the “selectivity” of adiponectin for AIS, since we did not test multiple pathologies, we cannot say that these levels are specific for AIS. However, like multiple other biomarkers, we believe that adiponectin is useful together with the clinical context and suspicion of stroke, and not for the general screening of the population. At least, with our current study we cannot explore this aspect as the design of the study was aimed at exploring adiponectin levels in the context of clinical suspicion of AIS and explore the relationship with clinical features of AIS.

  1. The level of adiponectin in the AIS group was highly variable. Have the authors considered evaluating the AIS patients by quartiles according to their serum adiponectin levels?

R: Thank you for this question. Indeed, the spectrum of adiponectin values in the AIS patients is a lot wider than that of the control group. However, this is not uncommon for various biomarkers. For example, CA125 which is used for the investigation of ovarian cancer patients can present values in the range of 10000UI, when the threshold for the normal population is 35UI/mL. We indeed thought of conducting an analysis based on the quartiles, but this division would result in too few patients in each group and negatively impact the statistical power of the tests. Thus, we chose to conduct classical analyses based on the clinical aspect of the patients.

  1. The discussion section includes extensive references to other studies, which sometimes makes it resemble a review. To enhance discussion, the authors could more clearly state how their results align or differ from previous reports and then discuss the clinical implications of these findings for the study population.

R: We agree that there are multiple references, and we discussed this aspect during the writing process. We feel that the works of other who explored similar issues should be acknowledged and thus decided to discuss their works in relationship with our results. Nevertheless, we rearranged the discussion section to highlight more our results. We thank you for pointing this out as it has allowed us to bring forward our results.

Reviewer 3 Report

Comments and Suggestions for Authors

This aimed to evaluate the diagnosis capacity 16 of adiponectin levels and its association with the clinical parameters of patients with acute ischemic stroke. I have few comments prior the acceptance for publication.

Abstract: well written

Introduction: what is the hypothesis of study?

Methods: What are the body composition data? Methods and tools for this?

Results: To add the body composition data.

Discussion: What is relationship between body composition, in particular adiposity with adiponectin?

Author Response

This aimed to evaluate the diagnosis capacity of adiponectin levels and its association with the clinical parameters of patients with acute ischemic stroke. I have few comments prior the acceptance for publication.

Abstract: well written

R: Thank you for taking the time to review our manuscript and for all the constructive comments and suggestions, please find below a point-by-point response

Introduction: what is the hypothesis of study?

R: Thank you for this question. We rephrased the aims of the study to put more highlights on the gaps we intended to address with the analysis: “Starting from the body of evidence connecting the adipose tissue, adipokines, and vascular disorders, this study aimed to investigate the diagnostic capacity of adiponec-tin in patients presenting at the emergency unit with AIS. The second aim was to evaluate the association of adiponectin levels with clinical parameters of AIS as well as the prognosis of these patients. Moreover, we assessed the evolution of adiponectin levels in the first two days from the AIS onset.”(lines 79-84)

Methods: What are the body composition data? Methods and tools for this? Results: To add the body composition data.

R: Body composition was assessed only through BMI. Obesity was defined as a BMI ≥ 30. Please find this information added in Table 2.

Discussion: What is relationship between body composition, in particular adiposity with adiponectin?

R: The BMI of the AIS patients and its association with adiponectin levels both at 24h and at 48h are presented in table 2. The results showed significantly lower levels of adiponectin in patients with obesity at both timepoints.